# Phenolic Compounds and Antioxidant Capacity of Sea Cucumber (*Cucumaria frondosa*) Processing Discards as Affected by High-Pressure Processing (HPP)

**DOI:** 10.3390/antiox11020337

**Published:** 2022-02-09

**Authors:** Abul Hossain, JuDong Yeo, Deepika Dave, Fereidoon Shahidi

**Affiliations:** 1Department of Biochemistry, Memorial University of Newfoundland, St. John’s, NL A1C 5S7, Canada; abulh@mun.ca (A.H.); jy2402@mun.ca (J.Y.); 2Department of Food Science and Biotechnology of Animal Resources, Konkuk University, Seoul 05029, Korea; 3Marine Bioprocessing Facility, Centre of Aquaculture and Seafood Development, Marine Institute, Memorial University, St. John’s, NL A1C 5R3, Canada

**Keywords:** *Cucumaria frondosa*, sea cucumber processing waste, high-pressure processing (HPP), phenolics, UHPLC-MS/MS

## Abstract

Sea cucumber processing discards, which include mainly internal organs, represent up to 50% of the sea cucumber biomass, and are a rich source of bioactive compounds, including phenolics. This work aimed to extract free, esterified, and insoluble-bound phenolics from the internal organs of the Atlantic sea cucumber (*C. frondosa*) using high-pressure processing (HPP) pre-treatment. The sea cucumber internal organs were subjected to HPP (6000 bar for 10 min), followed by the extraction and characterization of phenolics. Samples were evaluated for their total contents of phenolics and flavonoids, as well as several in vitro methods of antioxidant activities, namely, free radical scavenging and metal chelation activities. Moreover, anti-tyrosinase and antiglycation properties, as well as inhibitory activities against LDL cholesterol oxidation and DNA damage, were examined. The results demonstrated that HPP pre-treatment had a significant effect on the extraction of phenolics, antioxidant properties, and other bioactivities. The phenolics in sea cucumber internal organs existed mainly in the free form, followed by the insoluble-bound and esterified fractions. Additionally, UHPLC-QTOF-MS/MS analysis identified and quantified 23 phenolic compounds from HPP-treated samples, mostly phenolic acids and flavonoids. Hence, this investigation provides fundamental information that helps to design the full utilization of the Atlantic sea cucumber species and the production of a multitude of value-added products.

## 1. Introduction

The sea cucumber is harvested mainly for food, and it is also used in traditional medicine, especially in the Asian culture. As a result, there has been a growing interest in identifying biologically active compounds in sea cucumbers and their potential health benefits. The sea cucumber is well known for demonstrating various biological and pharmacological properties, including antioxidant, anticancer, anticoagulant, antithrombotic, and antimicrobial effects [1]. The components responsible for biological activities are glycosaminoglycans, triterpene glycosides (saponins), chondroitin sulfate, fucoidans, proteins (peptides), and other secondary metabolites, including phenolics [2,3]. For instance, Pranweerapaiboon et al. [4] reported that the phenolic-rich ethyl acetate fraction of sea cucumber (*H. scabra*) extracts inhibited the synthesis of pro-inflammatory cytokines, mainly inducible nitric oxide synthase (iNOS), nitric oxide (NO), tumor necrosis factor-α (TNF-α), interleukin-1β (IL-1β), and prostaglandin E2 (PGE2).

The orange-footed sea cucumber (*C. frondosa*) is abundant in the North Atlantic Ocean. This echinoderm species has contributed significantly since 2012 to the seafood industries in Newfoundland and Labrador (NL), Canada, when it was upgraded to a commercial status from its emerging state. According to the Department of Fisheries and Oceans (DFO) of Canada, the landed amount of sea cucumber has gradually increased in NL: it was 6.7 thousand MT and valued at USD 8.1 million in 2020. However, only limited information is available on the harvesting and potential utilization of sea cucumbers, and this hinders the expansion of this highly marketable species. Moreover, the body wall (representing around 50% of the total body weight) of this species is the major marketable product, which is sold mainly in the dried form. However, during sea cucumber processing, non-marketable portions, such as internal organs, including the gonads, respiratory tract, and intestines, are ultimately discarded as processing waste. These visceral by-products represent up to 50% of the harvest biomass and are a rich source of fatty acids, amino acids, vitamins, and minerals, as well as carotenoids and phenolics [5,6]. Various species of sea cucumber have different levels of phenolic compounds in different body parts, with varied antioxidant properties. This could be due to the existing difference in geographic locations, food habits, and harvest time. Therefore, suspension-feeding species (e.g., *C. frondosa*) may have more phenolics compared to deposit-feeding species.

Phenolic compounds serve as antioxidants, and have been used for improving the oxidative stability of foods and preventing oxidative stress in the human body. These antioxidants can hinder free radical chain reactions by donating an electron or a hydrogen atom to free radicals. Several studies have shown the therapeutic effect of phenolic compounds, such as for the inhibition of the formation of advanced glycation end products (AGEs), low-density lipoprotein (LDL) oxidation, DNA oxidation, tyrosinase and α-glucosidase inhibition activity, and hyperglycemic properties, among others. In addition, phenolic compounds can also control the oxidation of meat, fish, or lipid-containing food products during storage, which can be monitored by evaluating their oxidation products [7,8,9,10]. Therefore, using extracts rich in phenolic compounds from natural sources in food and medicine has become increasingly popular in recent years. Based on their solubility in the extraction medium, phenolic compounds can be divided into two groups of soluble (free and esterified/etherified) and insoluble-bound forms. Usually, free and esterified phenolics are trapped by weak interaction with other compounds in the seed vacuole, whereas insoluble-bound phenolics are bound through covalent bonds in the cell wall matrices of plant [11]. Hence, special techniques need to be followed to extract insoluble-bound phenolics.

The extraction of phenolic compounds using high-pressure processing (HPP) has attracted increased attention as an effective pre-treatment method. Several studies have stated that HPP increases the content of phenolic compounds and the antioxidant activity of foods. For example, da Silveira et al. [12] reported that HPP treatment (500 MPa for 5 min) increased the total amount of phenolic compounds and the antioxidant activity of açaí (*Euterpe oleracea*) juice and preserved the anthocyanins up to 40% better compared to thermal pasteurization (85 °C for 1 min). This could be due to the improvement of solvent penetration into samples through the disruption of cell membranes by HPP, which may increase the mass transfer and permeability and, therefore, improve the extraction [13]. Moreover, HPP could inactivate the residual enzymatic activity of polyphenol oxidase (PPO) and peroxidase (POD) in foods, which are linked to the enzymatic degradation of anthocyanins, causing browning [12]. Therefore, HPP could be applied as an alternative process to increase the extraction efficacy of bioactive compounds (e.g., phenolic compounds). However, no information is available on the use of HPP in any species of sea cucumber. Moreover, previous studies have examined the phenolics of the body wall from different species of sea cucumber, including *C. frondosa* [14]. To date, no attempt has been made to determine and identify phenolic compounds from any species of sea cucumber processing discards, which represent almost half of the sea cucumber body weight. Therefore, the aim of this study was to investigate soluble and insoluble-bound phenolics present in Atlantic sea cucumber processing waste using HPP pre-treatment, and to analyze their corresponding antioxidant activities.

## 2. Materials and Methods

### 2.1. Sample Collection and Procurement Materials

Atlantic sea cucumbers (*C. frondosa*) were collected from Newfoundland, Canada. The compounds 2,2′-azobis(2-methylpropionamidine) dihydrochloride (AAPH), 2,2′-azino-bis(3-ethylbenzothiazoline-6-sulfonic acid) (ABTS), 5,5-dimethyl-l-pyrroline *N*-oxide (DMPO), 2,2-diphenyl-1-picrylhydrazyl (DPPH), *p*-nitrophenyl-α-D-glucopyranoside (PNPG), supercoiled plasmid pBR322 DNA, human LDL cholesterol, tyrosinase, and phenolic standards were purchased from Sigma-Aldrich Ltd. (Oakville, ON, Canada). All other chemicals and solvents were of an analytical or chromatographic grade, and were obtained from Sigma-Aldrich Ltd. (Oakville, ON, Canada) and Fisher Scientific Co. (Nepean, ON, Canada).

### 2.2. Sample Preparation

The fresh sea cucumbers were shipped to the food processing pilot plant (Marine Institute, St. John’s, NL, Canada) and graded based on their weight (180–200 g). The internal organs (gonads, respiratory tracts, and intestines) were manually separated using a knife and placed in Ziploc bags for HPP pre-treatment. An initial experiment was performed to optimize HPP pressure (2000, 4000, and 6000 bar) and time (5, 10, and 15 min) based on the total phenolic content and antioxidant activity (data not shown). After that, an optimum parameter (6000 bar for 10 min) was applied to conduct the HPP pre-treatment for the current study. For that, the sealed samples were placed into pilot-scale NC Wave 6000/55 HPP equipment (Hyperbaric, Burgos, Spain) using HPP vessels. Water was used as a pressure-transmitting medium at an ambient temperature (20 °C). Finally, samples were taken out from the vessels after releasing the pressure quickly. However, untreated sea cucumber internal organs were kept without any treatment in a −18 °C freezer for further analysis. After freezing, both untreated and HPP-treated samples were freeze-dried (Labconco, Kansas City, MO, USA) for 72 h and then ground to obtain a powder with a particle size of ≤1 mm. After sieving, samples were defatted with hexane (1:5 *w*/*v*) and stored at −18 °C for further analysis. Figure 1 represents the experimental design for evaluating the antioxidant potential of phenolics extracted from Atlantic sea cucumber processing discards.

### 2.3. Extraction of Free, Esterified, and Insoluble-Bound Phenolics

The free and insoluble-bound phenolics of sea cucumber internal organs were extracted by using ultrasound-assisted extraction and alkaline hydrolysis, respectively, as described by Ambigaipalan, de Camargo, and Shahidi [15], with slight modification. For free phenolics, HPP-treated and untreated samples were placed in an ultrasonic water bath (300 Ultrasonik, Rancho Cucamonga, CA, USA) at 30 °C for 20 min after mixing with 70% acetone (1:10, *v/v*). The extraction was repeated three more times, and the resulting slurry was centrifuged at 5000× *g* for 6 min (Thermo Scientific™ Sorvall LYNX 6000 Superspeed Centrifuge, Thermo Fisher Scientific, Pittsburgh, PA, USA). After removing the solvent from the combined supernatants using a rotary evaporator (R-300, Buchi, Flawil, Switzerland), the aqueous suspension was acidified (pH 2.0) with 6 M HCl. After that, free phenolics were extracted with ethyl acetate/diethyl ether (1:1, *v/v*) in a separatory funnel. The extraction was repeated four more times, and the organic phase was evaporated and then dissolved in 10 mL 80% HPLC-grade methanol. For esterified phenolics, the remaining water fraction from the free phenolic fraction was hydrolyzed with an equal volume of 4 M sodium hydroxide for 4 h. The suspension was acidified and then centrifuged and evaporated as detailed above. Moreover, insoluble-bound phenolics were extracted from the meal (solid) residue obtained from the centrifugation of free phenolics. For that, samples were hydrolyzed using 4 M sodium hydroxide followed by acidification and evaporation, as described above. Finally, all extracts in 80% HPLC-grade methanol were stored at −18 °C for further analysis of the total phenolics, individual phenolic profiles, and the antioxidant activity of sea cucumbers.

### 2.4. Total Phenolic Content (TPC)

The total phenolic contents of the sea cucumber (treated and untreated) extracts were evaluated using the procedure explained by Singleton and Rossi [16], with some adjustments. For this, 0.5 mL of extracts was added to 0.5 mL of 10% Folin–Ciocalteu’s phenol reagent, and then neutralized by the addition of 1 mL 7% sodium carbonate. After adding 8 mL of distilled water, the mixture was kept in the dark for 2 h, and the absorbance was subsequently read at 760 nm using a UV–visible spectrophotometer (HP 8452 A diode array spectrophotometer, Agilent Technologies, Palo Alto, CA, USA). The results were calculated and reported as milligrams of gallic acid equivalents (GAE) per 100 g sample.

### 2.5. Total Flavonoid Content (TFC)

The total flavonoid contents were measured using the method of Yeo and Shahidi [17], with slight modification. Briefly, 1 mL extract was added to 4 mL of distilled water, and then 0.3 mL of a 5% sodium nitrite solution was added and kept for 5 min. After that, the mixtures were allowed to react with 0.3 mL of 10% aluminum chloride, 2 mL of 1 M sodium hydroxide, and 2.4 mL of distilled water. After 15 min, the absorbance was measured at 510 nm using a UV–visible spectrophotometer, as mentioned above, and the content was expressed as mg catechin equivalents (CE) per 100 g sample.

### 2.6. ABTS Radical Scavenging Activity

The ABTS radical scavenging activity was evaluated as described by Hossain, Moon, and Kim [18], with some modifications. Concisely, 2.6 mM potassium persulfate solution and 7.4 mM ABTS solution were mixed in equal volumes and kept in the dark. After 12 h, the ABTS stock solution was diluted with ethanol (ABTS: ethanol; 1:24) to acquire an absorbance of 1.5 ± 0.02 at 734 nm. Finally, 0.3 mL of phenolic extract was added with 2.7 mL of the ABTS solution, and the absorbance was measured after 10 min at the same wavelength as mentioned above. The results were presented as milligrams of Trolox equivalents (TE) per 100 g sample.

### 2.7. DPPH Radical Scavenging Activity

The DPPH radical scavenging activity was measured based on the method described by Ambigaipalan, de Camargo, and Shahidi [15], with some modifications. Briefly, 0.1 mL of phenolic extract in methanol was added with 3.9 mL of a methanolic DPPH solution (0.30 mM). After keeping for 30 min in the dark, the absorbance was read using a Bruker-E-scan electron paramagnetic resonance (EPR) spectrometer (Bruker E-scan, Bruker BioSpin Co., Billerica, MA, USA). The activity was expressed as milligrams of Trolox equivalents (TE) per 100 g sample.

### 2.8. Hydroxyl Radical Scavenging Activity

The hydroxyl radical scavenging activity was measured using the procedure of Chandrasekara and Shahidi [19]. For that, samples were dissolved in 75 mM phosphate buffer (pH 7.2) after removing the methanol from the sea cucumber extracts. After that, the extracts (200 μL) were mixed with FeSO_4_ (200 μL, 10 mM), DMPO (400 μL, 17.6 mM), and hydrogen peroxide (200 μL, 10 mM). After 3 min, samples were passed through the sample cavity of a Bruker-E-scan EPR, and their spectra were recorded. The activity of the phenolic extracts was expressed as milligrams of Trolox equivalents (TE) per 100 g sample.

### 2.9. Metal Chelation Activity

The metal chelation activity was determined according to the procedure of Ambigaipalan, de Camargo, and Shahidi [15], with some modifications. Firstly, 0.5 mL of 0.2 mM FeCl_2_, 0.2 mL of 5 mM ferrozine, and 2.9 mL distilled water was added to the extract. After 12 min, the absorbance was read at 562 nm using a spectrophotometer, and the results were expressed as milligrams of EDTA (ethylenediaminetetraacetic acid) equivalents per 100 g sample.

### 2.10. Cupric Ion-Induced Human Low-Density Lipoprotein (LDL) Peroxidation

The LDL oxidation inhibition activity of sea cucumbers was measured based on the method of Ambigaipalan and Shahidi [20], with slight modification. Briefly, human LDL (5 mg/mL) was dialyzed in 10 mM phosphate buffer (PBS, pH 7.4, 0.15 M NaCl) at 4 °C for 12 h. Then, 0.8 mL of diluted LDL cholesterol (0.04 mg LDL/mL) was mixed with 0.1 mL sea cucumber extract (0.1 mg/mL), followed by pre-incubation at 37 °C for 15 min. To initiate the oxidation reaction, a solution of cupric sulfate (0.1 mL, 100 μM) was added to the mixture, followed by incubation at 37 °C for 22 h. A diode array spectrophotometer was used to measure the formation of conjugated dienes (CD) produced from the oxidation of LDL cholesterol at 234 nm. Blanks were prepared by replacing LDL and CuSO_4_ with PBS.

### 2.11. Peroxyl and Hydroxyl Radical-Induced Supercoiled DNA Strand Scission

The DNA oxidation inhibitory activity of sea cucumber phenolic extracts was determined according to the method of Rahman, de Camargo, and Shahidi [21], with slight modification. At first, supercoiled pBR322 plasmid DNA (50 μg/mL in PBS, 2 μL), PBS (2 μL), AAPH (7 mM, 4 μL), and phenolic extracts (0.1 mg/mL, 2 μL) were mixed to determine their inhibitory activity against peroxyl radical-induced oxidation. Similarly, DNA (50 μg/mL, 2 μL), H_2_O_2_ (0.5 mM, 2 μL), FeSO_4_ (0.5 mM, 2 μL), PBS (0.1 M, 2 μL), and phenolic extracts (6 mg/mL, 2 μL) were added to produce hydroxyl radicals. A blank (DNA and PBS) and a control (DNA, PBS, and free radical-generating reagents, such as FeSO_4_, H_2_O_2_, or AAPH) were prepared for each set of tests. Then, 1 μL of the loading dye, consisting of 0.25% xylene cyanol, 0.25% bromophenol blue, and 50% glycerol, was added to the reaction mixture after incubation at 37 °C for 1 h. The mixture (10 μL) was electrophoresed using a 0.7% agarose gel prepared in Tris-acetic acid-EDTA (TAE) buffer (40 mM Tris-acetate, 1 mM EDTA, pH 8.5), followed by the addition of SYBR safe (5 μL) to an agarose gel solution (50 mL) as a gel stain. A horizontal submarine gel electrophoresis apparatus (Owl Separation Systems Inc., Portsmouth, NH, USA) was used at 80 V for 90 min. The intensity (area %) of bands was calculated using Alpha-Imager gel documentation (Cell Biosciences, Santa Clara, CA, USA). The inhibitory effect of the phenolic extracts was evaluated using the following equation:DNA retention (%) = (Area of supercoiled DNA with oxidative radical and extract/Area of supercoiled DNA in control) × 100(1)

### 2.12. Anti-Tyrosinase Activity

The anti-tyrosinase activity of extracts was evaluated using the method explained by Muddathir et al. [22], with slight modification. For that, phenolic extracts (70 µL) were added to tyrosinase (30 µL, 333 unit/mL in phosphate buffer 50 mM, pH 6.5) and incubated at 37 °C for 5 min. After that, 110 μL of substrate (L-tyrosine 2 mM) was added and incubated for 30 min. The sample solution was placed in the 96-well plate, and the absorbance was read using a microplate reader (Gen5™ Microplate Reader, BioTek Instruments, Winooski, VT, USA) at 510 nm. A blank was prepared using a sample solution of all the reaction reagents without enzymes, and a positive control was made with kojic acid (anti-tyrosinase agent).

### 2.13. Antiglycation Activity

The antiglycation activity of sea cucumber phenolic extracts was determined using the method described by Hu, Wang, and Shahidi [23], with some modifications. Advanced glycation end products (AGEs) form through a non-enzymatic reaction, which can be measured by analyzing their fluorescence intensity. Briefly, BSA (2 mg/mL, 500 µL) was incubated with phenolic extracts (100 µL) in PBS (0.1 M, pH 7) and D-glucose (33 mM, 400 µL) at 37 °C for 7 days. A control was prepared using PBS, while the positive control was made using 1 mM aminoguanidine (a typical inhibition agent of AGEs). After that, the sample solution (100 µL) was placed in the 96-well plate, and the level of AGEs was evaluated by monitoring fluorescence intensity using a microplate reader. The excitation and emission wavelengths were 355 and 405 nm, respectively.

### 2.14. UHPLC-QTOF-MS/MS Analysis

Ultra-high-performance liquid chromatography with quadrupole time of fight and mass spectrometer (UHPLC-QTOF-MS/MS) analysis was carried out as described by Ambigaipalan, de Camargo, and Shahidi [15], with some modifications. The identification and quantification of the phenolic compounds of sea cucumber were conducted using an Agilent 1290 UHPLC system equipped with an autosampler (G4226A), a binary pump (G4220A), and a system controller linked to a OpenLab software (Agilent Technologies, Palo Alto, CA, USA). A Synergi™ Fusion LC-18 column (50 × 2 mm, 4 µm) was used for separation, and a diode array detector (DAD, G4212A) at 280 nm was applied for detection. The mobile phase was composed of 0.1% formic acid in water (eluent A) and methanol (eluent B). The elution gradient was at 0 min, 100% A; 5 min, 90% A; 35 min, 85% A; 45 min, 60% A; 50 min, 60% A; and 55–65 min, 100% A. The extracts were filtered using a 0.2 μm syringe filter, and the injection volume and flow rate were 5.0 µL and 0.4 mL/min, respectively. Moreover, UHPLC-MS/MS analysis was performed using a triple TOF 5600 system (AB SCIEX, Redwood City, CA, USA) with the electrospray ionization (ESI) ionization source in the negative ion mode. The mass spectrometer scanned the ions in the m/z range of 100 to 2000. The data were acquired and analyzed with PeakView^®^ software (AB SCIEX, Redwood City, CA, USA). The quasi-molecular weights, mass errors, and isotope patterns were obtained through the XIC manager tool in the Master View^®^ (AB SCIEX, Redwood City, CA, USA) software.

### 2.15. Statistical Analysis

Statistical analysis was performed among the different types of phenolics and HPP treatments. All tests were carried out in triplicate, and the data were presented as mean ± standard deviation. The data were processed by using a one-way analysis of variance (ANOVA), and mean separations were analyzed with Tukey’s HSD test using the IBM SPSS 27.0 for Windows (SPSS Inc., Chicago, IL, USA).

## 3. Results and Discussions

### 3.1. Total Phenolic Content (TPC) and Total Flavonoid Content (TFC)

The total phenolic and flavonoid contents of three different phenolic fractions from the untreated and HPP-treated sea cucumber extracts are presented in Figure 2. Among all extracts, free phenolics made the highest contribution to the TPC, followed by insoluble-bound and esterified phenolic fractions, regardless of HPP treatment. However, HPP pre-treatment significantly (*p* < 0.05) enhanced the TPC of all three different phenolic fractions, especially those of the free phenolics. The TPC ranged from 15.53 to 171.86 mg GAE/100 g for untreated samples, and 18.87 to 227.87 mg GAE/100 g for HPP-treated internal organs. Similarly, the TFC of the free, esterified, and insoluble-bound phenolic fractions from the HPP-treated samples were around 1.23, 1.5, and 1.18 times higher than those of their untreated counterparts, respectively. These results were in agreement with the findings of many previous studies, such as that of Andrés, Villanueva, and Tenorio [24], who stated that HPP significantly enhanced the extractability of phenolics from smoothies compared to untreated counterparts. This increase in TPC and TFC may be due to the disruption of the cell membranes following HPP, resulting in the availability of more phenolic compounds. Moreover, due to HPP, solvents may easily penetrate into samples and enhance the mass transfer and permeability, thus improving extraction [25]. Altuner et al. [26] showed that the content of phenolics of the *Maclura pomifera* fruits was significantly higher at 500 MPa than that at 250 MPa, and this could be due to the improvement of the solubility of phenolics based on the phase behavior theory upon the administration of HPP. Furthermore, Zhou et al. [27] found that HPP pre-treatment (500 MPa for 10 min) significantly improved the TPC of oil palm (*Elaeis guineensis*) fruit in all three phenolic fractions, mainly those of insoluble-bound phenolic fraction. The same study also reported that HPP had a greater effect on the TFC of the free phenolic fraction than the other fractions, as the free phenolics could partially be reversibly complexed with some plant matrix components, including cell walls, which could dissociate upon HPP and improve the extraction process. Nevertheless, in our study, HPP had a better effect on the free phenolic fraction of TPC and TFC, and this could be due to the presence of a very high amount of free phenolics in the sea cucumber internal organs, which could easily be liberated upon HPP. Notably, the TPC of HPP-treated free phenolic fraction (227.87 mg GAE/100 g) was higher than the sum of the TPCs in the esterified and insoluble-bound phenolic fractions (74.95 mg GAE/100 g), suggesting that the phenolics of sea cucumber internal organs occur mainly in the free form.

Several studies on the phenolics of different species of sea cucumbers, mainly focusing on the dried body wall, have reported that this echinoderm contains a significant amount of phenolics. However, various species of sea cucumber have different levels of phenolic compounds and varied antioxidant activities. For instance, Althunibat et al. [28] compared the phenolics of three Malaysian sea cucumber species (*Holothuria leucospilota, Holothuria scabra*, and *Stichopus chloronotus*), and reported that the extracts of *H. leucospilota* had a higher TPC (9.7 mg GAE/g) than *H. scabra* (1.53 mg GAE/g). On the other hand, Mamelona et al. [29] found a higher TPC in the Atlantic sea cucumber digestive tract (236 mg GAE/100 g), respiratory apparatus (200.1 mg GAE/100 g), muscles (194.1 mg GAE/100 g), and gonads (130.2 mg GAE/100 g) in the acetonitrile-rich fractions than water-rich fractions. They also reported that the TFC was higher in the gonads (59.8 mg rutin equivalents (RE)/100 g), followed by the digestive tract (44.1 mg RE/100 g), muscles (21.8 mg RE/100 g), and respiratory apparatus (9.6 mg RE/100 g). In our study, the TPC (free + esterified + insoluble-bound phenolics) in the HPP-treated internal organs (gonads + digestive tract + respiratory apparatus) was 302.82 mg GAE/100 g, which were higher compared to the untreated samples with 232.67 mg GAE/100 g. Similarly, the TFC (free + esterified + insoluble-bound phenolics) was 124.42 and 101.04 mg CE/100 g for HPP-treated and untreated samples, respectively. In another study, Zhong et al. [14] found that fresh Atlantic sea cucumbers with and without internal organs had a higher TPC than their dried counterparts, thus indicating the loss of phenolics upon drying. The Atlantic sea cucumber is a suspension-feeding species; hence, the phenolics in *C. frondosa* originate from the presence of phenolic-rich material in their diet. The sea cucumber may absorb phenolic compounds from an algae and/or microalgae origin, including phytoplankton and particles obtained from degrading marine macroalgae, which are abundant in phenolics such as anthocyanins, anthocyanidins, flavonoids, and tannins [14,29]. These phenolics may be involved in protecting sea cucumber against oxidative stress.

### 3.2. Antioxidant Activity

In vitro antioxidant properties, such as radical scavenging properties (DPPH, ABTS^•+^, and hydroxyl radical [HO^•^]) and the metal chelation of phenolic extracts from sea cucumber internal organs, were evaluated (Table 1). ABTS radicals can be reduced in the presence of hydrogen-donating antioxidants, and develop a blue-green chromophore, which can be measured at 734 nm [15]. In this study, ABTS radical scavenging activity was reduced significantly (*p* < 0.05) upon HPP, regardless of the phenolic fraction tested. For example, the activity of untreated internal organs in the free, esterified, and insoluble-bound fractions was 589.18, 89.67, and 109.03 mg TE/100 g, respectively, while those for the HPP-treated extracts were reduced to 565.43, 76.54, and 94.76 mg TE/100 g, respectively. This might be due to the partial activity of PPO and POD, leading to enzymatic degradation [12]. This finding is similar to that reported by De Ancos et al. [30], who found a reduction in ABTS radical scavenging activity upon HPP, possibly due to the oxidation of some bioactive compounds, including phenolics. The ABTS radical scavenging activity of both treated and untreated samples was well correlated with their total phenolic contents. Mention should be made that the ABTS radical scavenging activity of the Atlantic sea cucumber is being reported here for the first time. However, Husni et al. [31] found a good correlation between phenolic-rich extracts of S. japonicus and the ABTS and DPPH radical scavenging activities.

The DPPH radical scavenging activity was evaluated to assess the hydrogen atom- or electron-donating ability of the phenolic compounds extracted from the sea cucumbers. Regardless of HPP treatment, all phenolic fractions exhibited strong DPPH radical scavenging activity. HPP treatment significantly increased DPPH radical scavenging activity, regardless of phenolic fraction involved. For instance, the total (free + esterified + insoluble-bound phenolics) DPPH radical scavenging activity in the HPP-treated samples was 462.96 mg TE/100 g, but it was reduced to 437.1 mg TE/100 g in the untreated counterparts. This finding gains support from studies of Zhou et al. [27], who showed that the DPPH radical scavenging activity of phenolics increased significantly upon HPP, regardless of the type of phenolics. Moreover, previous studies have reported that HPP significantly increases the antioxidant activity, which is perhaps related to the higher extractability of the antioxidant components (e.g., phenolics) [12,32]. Therefore, the increase in the DPPH radical scavenging activity of sea cucumber phenolics could be explained by the increase in the content and release of certain phenolic compounds upon HPP. Like ABTS radical scavenging activity, the highest level of DPPH radical scavenging activity was observed in the free phenolic fraction, followed by the insoluble-bound and esterified phenolic fractions, which is aligned with our TPC and TFC data. However, the scavenging effect was lower for the DPPH than the ABTS radical under the same conditions. Hossain et al. [33] reported that DPPH radical scavenging effect was lower when compared with the ABTS radical from natural extracts. A strong correlation existed between the DPPH radical scavenging activity and TPC, suggesting that the phenolics of sea cucumber internal organs have excellent free radical scavenging power. Similarly, Zhong et al. [14] determined the DPPH radical scavenging activity of Atlantic sea cucumber phenolics, and found that it has a strong correlation with TPC. The DPPH radical scavenging effect of phenolic extracts has also been evaluated from other sea cucumber species, and has been found to have potent antioxidant properties [28,31].

The hydroxyl radical is highly reactive, and is generated via a Fenton reaction and forms an adduct with DMPO. This adduct formation is decreased in the presence of antioxidants, which could be measured using EPR. All phenolic fractions exhibited strong hydroxyl radical scavenging properties, regardless of HPP treatment, in our study. However, HPP induced irregular changes in the hydroxyl radical scavenging activity. For example, HPP did not have any effect in the free and esterified phenolic fractions, whereas the HPP-treated insoluble-bound phenolic fraction had higher activity than their untreated counterparts. These results agree with those found in cashew apple juice, where Queiroz et al. [34] reported that HPP did not change the observed antioxidant activity. This could be due to the fact that the content of certain phenolic compounds that showed hydroxyl radical scavenging activity in the free and esterified phenolic fractions remain unchanged upon HPP. Moreover, the free phenolic fraction showed higher activity (~599 mg TE/100 g) than the other phenolic fractions. This is the first study of hydroxyl radical scavenging activity for any species of sea cucumber; thus, a direct comparison with the literature data is not possible. However, Mamelona et al. [29] determined the oxygen radical absorbance capacity (ORAC) of the internal organs of Atlantic sea cucumbers and found that the ORAC values for gonads were significantly correlated with TPC.

A polyphenolic compound becomes a chelator once it forms a coordinate complex with metal ions (e.g., Fe and Cu) and makes them inaccessible for involvement in the oxidation process. Therefore, metal chelation activity can be evaluated when a complex is formed between antioxidants and metal ions, resulting in the loss of intensity of ferrozine–ferrous color complex (pink color), which can be measured at 562 nm. HPP significantly improved the metal chelation activity in soluble phenolics in the present study, while no change was observed in the insoluble-bound fraction. This might be due to the higher extraction yield of HPP for soluble phenolics, causing higher antioxidant activity, as proposed by other authors [24,25]. For instance, De Ancos et al. [30] found that HPP treatment (at 200 and 400 MPa) significantly increased the FRAP value of orange juice phenolics. The highest activity was observed in the free phenolic fraction, followed by the esterified and insoluble-bound fractions. The unchanged activity of treated versus untreated samples in the insoluble-bound phenolic fraction could be linked to the amount and type of phenolics present in that particular fraction. It should be noted that, as of yet, metal chelation activity has not been determined for any species of sea cucumber.

Considering the different evaluation methods, the results indicate that antioxidant activities depend on the method employed, possibly due to the different mechanisms involved in various assays. Hence, HPP positively affected the metal chelation and DPPH radical scavenging activities, while nearly no effect was observed in the hydroxyl radical scavenging activity. Therefore, it can be concluded that HPP preserves the antioxidant activity of sea cucumber phenolics. Furthermore, the results of these assays indicate that sea cucumber internal organs contain some active antioxidant components, including phenolics. However, HPP significantly reduced the ABTS radical scavenging activity, possibly due to the mechanism of action involved in the ABTS assay or the oxidation of certain phenolic compounds. In addition, the Folin–Ciocalteu method may measure other non-phenolic components, such as proteins, which could increase the TPC, but not improve the antioxidant properties. Consequently, at least two different assays should be used to evaluate the antioxidant effects of natural extracts for better reliability [35].

### 3.3. Cupric Ion-Induced Human Low-Density Lipoprotein (LDL) Peroxidation

It is believed that the oxidation of LDL plays a key role in coronary heart disease and atherogenesis in humans. This could occur via the narrowing and/or blocking of blood vessels due to the oxidation of LDL. The oxidation could start with the action of metal ions (e.g., Cu^2+^ and Fe^2+^) or reactive oxygen species (ROS) [19]. In this study, copper sulfate was used to initiate LDL oxidation, and the susceptibility of oxidation was spectrophotometrically monitored by measuring conjugated dienes (CD) as primary oxidation products at 234 nm. The absorbance of samples was stabilized after 8 h, while the absorbance increased gradually for the control (LDL cholesterol without phenolics) during the incubation period, suggesting the development of oxidation products, mostly CD. No significant (*p* > 0.05) difference was found between untreated and HPP-treated samples, except in the free phenolic fraction, where HPP increased the inhibitory activity (Figure 3). This could be linked to the hydroxyl radical scavenging activity in the esterified phenolic fraction and the metal chelation activity in the insoluble-bound phenolic fraction, where no significant difference was found between the untreated and HPP-treated samples. The highest inhibitory activity was obtained in the free phenolic fraction, followed by esterified and insoluble-bound phenolics. Notably, the esterified phenolic fraction showed a higher level of inhibition than the insoluble-bound phenolic fraction, even though the TPC was greater in the insoluble-bound phenolic fraction. This could be due to the availability of certain phenolic compounds (e.g., *p*-hydroxybenzoic, Table 2) in the esterified phenolic fraction, which demonstrates a higher level of inhibitory activity against LDL oxidation. Additionally, LDL oxidation inhibition was moderately correlated (r^2^ = 0.61) with TPC, suggesting that the inhibitory activity of sea cucumbers depended on the specific type of phenolic compounds and their chemical structures. It has been reported that phenolic compounds have the potential to inhibit LDL cholesterol oxidation, and the possible pathways could be related to the scavenging of free radicals, as well as the chelation of metal ions [15,21]. However, the inhibition of LDL cholesterol oxidation has not previously been determined from any species of sea cucumber phenolics.

### 3.4. DNA Strand Scission Inhibition Activity Induced by Hydroxyl and Peroxyl Radicals

DNA damage plays a key role in mutation and carcinogenesis in humans. ROS are continuously generated at the mitochondria via Fenton’s reaction and are involved in cell metabolism. For instance, hydroxyl radicals can be produced from the degradation of hydrogen peroxide and the interaction between hydrogen peroxide and superoxide. However, a higher level of ROS may lead to oxidative stress, which is responsible for the oxidation of DNA [36]. Thus, preventing peroxyl- and hydroxyl radical-induced DNA scission is crucial for avoiding cell damage. This study used supercoiled plasmid DNA to monitor the inhibitory activity of phenolic compounds against free radical-induced DNA strand scission. In the agarose gel electrophoresis, the supercoiled DNA could change its shape to nicked open circular and linear forms due to the oxidation of DNA induced by free radicals. Generally, the linear/nicked open form of DNA (oxidized DNA) moves more slowly through an agarose gel network in comparison with the supercoiled DNA (intact DNA), and the area of these bands can be used to calculate the inhibition percentage. This study used AAPH to generate peroxyl radicals and Fe^2+^ to produce hydroxyl radicals. The results suggest that all three phenolic fractions displayed strong inhibitory activity against peroxyl radical- and hydroxyl radical-induced DNA scission, regardless of HPP treatment (Figure 4). Nevertheless, no significant (*p* > 0.05) difference was observed between HPP-treated and untreated internal organs (data not shown). The highest inhibitory activity was obtained in the free (90.36%) phenolic fraction, followed by insoluble-bound (80.48%) and esterified (73.64%) phenolic fractions, against peroxyl radicals. Although the esterified phenolic fraction was not as efficient as the free and insoluble-bound counterparts with regard to the TPC and antioxidant activities, it demonstrated strong inhibitory activity against DNA damage. Therefore, the inhibitory activity does not depend fully on the quantity of phenolics, but rather on the type of phenolic compound(s) present in a particular phenolic fraction. However, the inhibitory activity against hydroxyl radicals was slightly lower than that against peroxyl radicals, which is aligned with the findings of Ambigaipalan, de Camargo, and Shahid [15], who found a similar trend for by-products of pomegranate phenolics. This could be related to the different half-life of each free radical, resulting in varied activities. Furthermore, a good correlation coefficient was observed between the inhibition of hydroxyl radical-induced DNA scission and hydroxyl radical scavenging activity, indicating that hydroxyl radicals play a key role in DNA damage. The DNA scission inhibitory activity has not yet been determined for phenolics of any sea cucumber species. However, phenolic compounds from various sources have been shown to inhibit the DNA oxidation induced by hydroxyl and peroxyl radicals [21,36]. The mechanism of DNA oxidation inhibition by phenolics is still unclear; however, possible pathways could be related to the scavenging and metal chelating activities of phenolics. Therefore, the phenolic compounds extracted from sea cucumber internal organs may serve as a good source of food ingredients for the prevention of DNA oxidation.

### 3.5. Anti-Tyrosinase Activity

Tyrosinase is a copper-containing oxidase that plays an important role in skin pigmentation. Tyrosinase forms dopachrome, thus stimulating the development of melanin pigments [37]. Usually, melanin is essential for preventing UV damage to skin, eyes, and hair; however, the overproduction of melanin is associated with neurodegenerative disorders and pigment variations. Apart from melanogenesis, tyrosinase is also involved in the browning reaction in some fruits and vegetables, and decreases their nutritive value [38]. Hence, the present study used L-tyrosine as a substrate to determine the inhibitory activity of mushroom tyrosinase. It was found that the HPP treatment significantly increased the tyrosinase enzyme inhibitory activity of phenolic extracts compared to their untreated counterparts (Figure 5). Especially, the free phenolic fraction in the HPP-treated samples showed higher activity (43.22%) compared to the untreated internal organs (30.76%), possibly due to the higher concentrations of phenolics in the free phenolic fraction of sea cucumber extracts. Moreover, kojic acid, a well-known tyrosinase inhibitor, was used as a positive control, which showed higher activity (86.23%) to L-tyrosine. The acquired data strengthen the results of the previous findings in the literature. For example, Zengin et al. [38] analyzed the tyrosinase inhibitory activity of *Ornithogalum narbonense*, and found that the phenolic-rich ethyl acetate extracts had a strong effect against the tyrosinase enzymes. Panzella and Napolitano [37] also show that flavonoids, mainly chalcones, as well as hydroxystilbenes, mostly resveratrol derivatives, play a prominent role as natural inhibitors against tyrosinase. Several inhibition mechanisms have been reported for tyrosinase inhibitors, among them the copper chelating activity of phenolic compounds is believed to play a main role. A good correlation was found between the TPC and tyrosinase inhibitory activity, suggesting that the phenolics of sea cucumber internal organs play a key role in inhibiting tyrosinase. Mention should be made that the tyrosinase inhibitory activity of the phenolics of sea cucumbers has not previously been reported.

### 3.6. Antiglycation Activity

Advanced glycation end products (AGEs) are mainly produced through a non-enzymatic Maillard reaction between the carbonyl group of reducing sugars and free amino groups in proteins. AGEs are generated in the advanced stage of the Maillard reaction when the intermediate components form an unstable Schiff base adduct, which rearranges to comparatively stable Amadori products [23]. AGEs are a group of complex and heterogeneous biomolecules that are linked to many ailments, including diabetes, cardiovascular disease, atherosclerosis, aging, and some types of cancer, when overproduced [39]. Consequently, the development of AGEs can be prevented by breaking sugar–protein cross-links, blocking sugar attachment, and scavenging the free radicals and carbonyls generated from the oxidation process. We found that HPP significantly increased the inhibitory activity against the formation of AGEs in the free and insoluble-bound phenolic fractions, while no changes were observed in the esterified phenolic fraction when compared to the untreated samples (Figure 6). For instance, the inhibitory activity of HPP-treated free, esterified, and insoluble-bound phenolic fractions was 68.23, 24.76, and 25.87%, respectively, while the values for their untreated counterparts were reduced to 48.17, 24.01, and 20.12%, respectively. This could be due to the higher content of phenolics present in the free and insoluble-bound phenolic fractions, mainly in the HPP-treated samples. Błaszczak et al. [40] also showed that HPP-treated phenolic-rich kiwiberry (*A. arguta*) extract had twice the antiglycation effect compared to its untreated counterparts. Nevertheless, aminoguanidine, a typical inhibitor of AGEs, demonstrated a higher inhibitory activity (80.65%) than the samples. Furthermore, the inhibitory activity against the formation of AGEs was positively correlated with TPC, suggesting that phenolic compounds had a great influence on the antiglycation effect. Previous studies have documented that phenolic-rich extracts have the potential to inhibit the formation of AGEs [41], perhaps due to their scavenging and chelating properties, and could trap reactive carbonyl species through adduct formation [42]. There are no similar studies in the literature for sea cucumber phenolic extracts in terms of antiglycation activity. Thus, the sea cucumber, with its phenolic compounds, may serve as a good source of food ingredients for minimizing the formation of AGEs.

### 3.7. Identification and Quantification of Phenolic Compounds

Phenolic compounds were identified using UHPLC-QTOF-MS/MS by comparing their retention times and the ion fragmentation patterns of authentic phenolic acid standards, including caffeic, gallic, sinapic, *p*-coumaric, *p*-hydroxybenzoic, cinnamic, protocatechuic, vanillic, ferulic, syringic, ellagic, and chlorogenic acids, as well as quercetin and catechin, under the same conditions as the tested samples. For example, compounds **1**–**7**, **11**, **13**–**15**, and **18**–**19** were identified by comparing their UV spectral data, retention times, and fragment ions with corresponding standard compounds (Table 2). Other compounds without standards were tentatively identified using UV spectral, mass spectral, and literature data. The compounds that were tentatively identified were homovanillic acid, hydroxygallic acid, *p*-coumaroyl glycolic acid, caffeoyl glucoside, chicoric acid, *p*-hydroxybenzaldehyde, *p*-hydroxycoumarin, scopoletin, and leachianol. In this study, **18**, **14**, and **16** phenolic compounds were identified in the free, esterified, and insoluble-bound phenolic fractions of HPP-treated internal organs, respectively. However, **14**, **11**, and **13** phenolics were identified in their untreated counterparts, respectively. Considering these results along with our TPC and TFC data, the contents of phenolic compounds were maximized when HPP was used as a pre-treatment, mainly in the free phenolic fraction. This could be due to the higher extraction yield from using HHP on sea cucumber internal organs. This finding is similar to that reported by Zhou et al. [27], who identified a higher number of phenolic compounds from the HPP-treated oil palm fruit than their untreated counterparts. Notably, a significant number of phenolic compounds were identified in the esterified phenolic fraction, though the TPC was lower in that particular fraction than the other two phenolic fractions.

**Table 2 antioxidants-11-00337-t002:** List of phenolic compounds identified in HPP-treated and untreated internal organs.

C#	Compounds	[M − H]^−^ (*m/z*)	RT (min)-UV	MS^2^ Ion Fragments	HPP-Treated			Untreated		
					F	E	IB	F	E	IB
**1**	*p*-Hydroxybenzoic acid *	137	3.52	121	p	p		p	p	
**2**	Cinnamic acid *	147	42.17	103, 131, 135	p	p	p	p		p
**3**	Protocatechuic acid *	153	2.5	109	p		p	p	p	p
**4**	*p*-Coumaric acid *	163	6.78	119	p	p	p	p	p	p
**5**	Vanillic acid *	167	4.56	105, 108, 121, 123	p	p	p		p	p
**6**	Gallic acid *	169	0.43	125	p	p	p	p	p	p
**7**	Caffeic acid *	179	5.15	135			p			p
**8**	Homovanillic acid	181	0.37	181, 137	p					
**9**	Hydroxygallic acid	187	13.5	125, 169	p	p	p	p	p	p
**10**	Isoferulic acid	193	46.69	133, 179			p			
**11**	Syringic acid *	197	5.36	109, 123, 153, 163, 179	p	p	p	p	p	p
**12**	*p*-Coumaroyl glycolic acid	221	42.29	147, 175, 179		p	p			
**13**	Sinapinic acid *	223	9.17	175, 179, 208	p	p		p		
**14**	Ellagic acid *	301	47.86	229, 257	p	p	p	p	p	p
**15**	Chlorogenic acid *	353	5.32	179, 191, 207	p		p	p		p
**16**	Caffeoyl glucoside	387	45.08	341	p			p		
**17**	Chicoric acid	473	44.2	311, 293, 219, 179, 135			p			
**18**	Catechin *	289	4.99	203, 245	p	p	p	p	p	p
**19**	Quercetin *	301	38.28	121, 151, 179, 255, 257, 273	p	p	p	p	p	p
**20**	*p*-Hydroxybenzaldehyde	121	5.06	92	p	p	p	p	p	p
**21**	*p*-Hydroxycoumarin	161	2.07	105, 121		p				
**22**	Scopoletin	191	43.15	147	p					
**23**	Leachianol F	471	41.65	121, 153, 287, 349, 453	p					
Total number of compounds					18	14	16	14	11	13

RT, retention time; F, free; E, esterified; IB, insoluble-bound. p Indicates the presence of the compound in the fraction. * Identified with authentic standard.

The quantification of phenolic compounds was accomplished using their corresponding standard derivatives. For compounds with no standards, quantification was carried out using their corresponding aglycones. The contents of phenolics in sea cucumber internal organs were expressed as mg/100 g of samples (Table 3). In the UHPLC analysis, the total content of phenolics (the sum of all compounds quantified from free, esterified, and insoluble-bound phenolics) was 81.06 and 62.5 mg/100 g in the HPP-treated and untreated internal organs, respectively. Therefore, HPP pre-treatment significantly increased the content of certain phenolic compounds, while other compounds remain unchanged. The literature reports support our results regarding the positive effects of HPP on the content of phenolic compounds [12]. For instance, Zhou et al. [27] reported that the contents of individual phenolic compounds were notably enhanced when oil palm fruits were treated with HPP. This could be due to the disruption of the cell walls upon HPP, causing a favorable condition to release phenolic compounds. Moreover, a similar flavonoid profile in untreated and HPP-treated orange juice was obtained by de Ancos et al. [30]. In this study, the major compounds were found to be cinnamic acid (3.24 mg/100 g), protocatechuic acid (3.69 mg/100 g), *p*-coumaric acid (2.89 mg/100 g), gallic acid (3.22 mg/100 g), hydroxygallic acid (3.25 mg/100 g), catechin (5.8 mg/100 g), and quercetin (3.05 mg/100 g) in the free phenolic fraction, protocatechuic acid (1.39 mg/100 g) and *p*-coumaric acid (1.38) in the esterified phenolic fraction, and *p*-coumaric acid (2.05 mg/100 g) and chlorogenic acid (3.06 mg/100 g) in the insoluble-bound phenolic fraction of HPP-treated internal organs. These compounds have been reported to show strong antioxidant activities, which may contribute to the in vitro biological activities of the tested extracts. Most phenolic compounds identified from the sea cucumber internal organs were phenolic acids and flavonoids. The TPC (F + E + IB) obtained from the UHPLC analysis was 82.06 mg/100 g in the HPP-treated internal organs, while the TPC observed from the Folin–Ciocalteu method was 302.82 mg GAE/100 g. The higher phenolics obtained using the Folin−Ciocalteu’s reagent could be related to the extraction of nonphenolic compounds, such as reducing sugars, amino acids, and other organic acids. Thus, individual phenolic profiles should be examined using reliable methods, such as UHPLC-MS.

However, this study has identified and quantified a total of 23 phenolic compounds in sea cucumber internal organs. To the best of our knowledge, 11 components were identified for the first time in any species of sea cumber. For example, Alper and Günes [43] identified epicatechin, 2,5-dihydroxybenzoic acid, ellagic acid, gallic acid, chlorogenic acid, 3,4-dihydroxybenzoic acid, *p*-hydroxybenzoic acid, vanillic acid, caffeic acid, *p*-coumaric acid, ferulic acid, cinnamic acid, rutin, naringin, and quercetin from the dried body wall of *Holothuria tubulosa*, and the most abundant component was epicatechin (790.091 µg/g). Moreover, Telahigue et al. [44] identified quinic acid, gallic acid, caffeic acid, syringic acid, trans-ferulic acid, o-coumaric acid, rosmarinic acid, and salvianolic acid from dried *Holothuria forskali*, and among them, quinic acid was abundant in the digestive tract, gonad, and respiratory tree. Additionally, phenolic compounds were identified and quantified from the *Holothuria atra* and *Holothuria arenicola*, and the major compounds were found to be chlorogenic acid (80.34–92.86%), pyrogallol, rutin, coumaric acid, and catechin [45,46]. However, this study identified and quantified the phenolic compounds from the Atlantic sea cucumber for the first time, and the variation in phenolics could be linked to the food habits, body parts, geographic locations, and harvesting times of sea cucumbers.

## 4. Conclusions

Atlantic sea cucumber internal organs contained a significant amount of phenolics that showed strong antioxidant activity. HPP pre-treatment served as an efficient treatment to extract phenolics and preserve antioxidant activity in the sea cucumber and its processing waste. Consequently, HPP pre-treatment could be used as an effective means to retain the antioxidant activities of phenolic-rich extracts from sea cucumbers. This systematic approach addresses both the environmental concerns and economic sustainability of the sea cucumber industries through a full utilization of sea cucumber processing discards. Moreover, the free phenolic fraction was the major groups of phenolics found in sea cucumbers, which mainly consisted of phenolic acids and flavonoids. The antioxidant efficacy of phenolic compounds revealed the potential of using sea cucumber discards rich in phenolic compounds as functional food ingredients and nutraceuticals for health promotion.

## Figures and Tables

**Figure 1 antioxidants-11-00337-f001:**
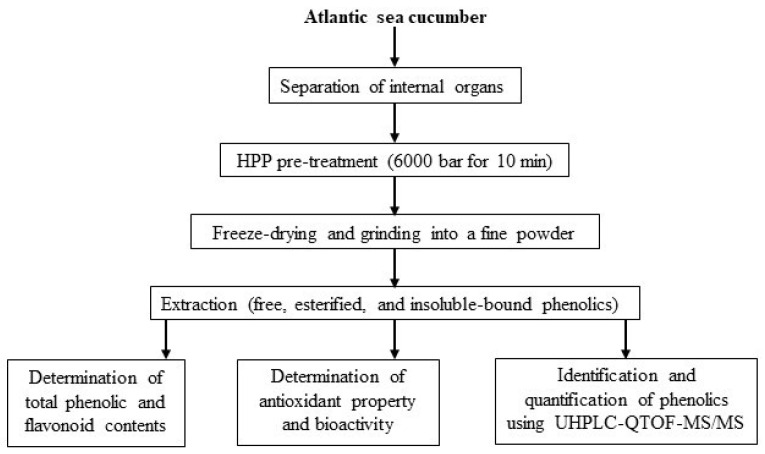
Experimental design for evaluating antioxidant potential of sea cucumber phenolics.

**Figure 2 antioxidants-11-00337-f002:**
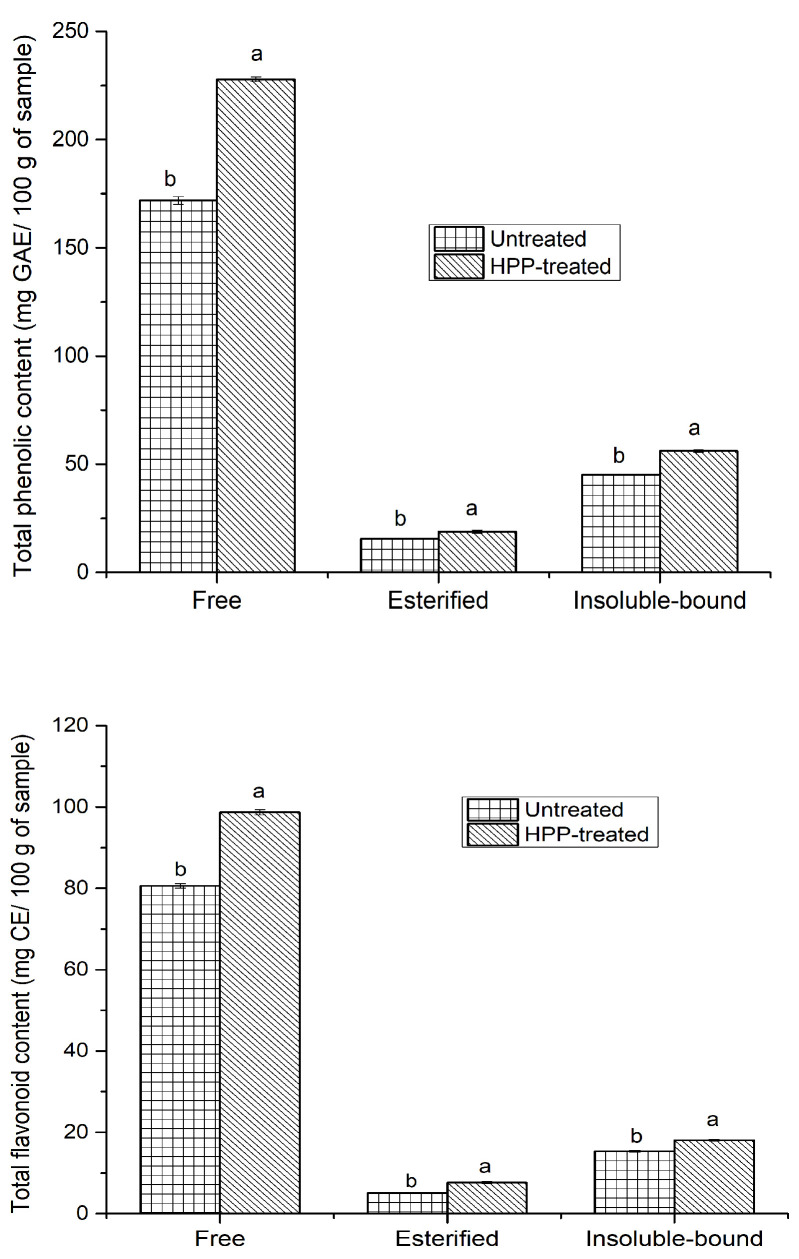
Total phenolic and flavonoid contents of HPP-treated and untreated sea cucumber in mg gallic acid and catechin equivalents per 100 g of sample, respectively. Different lowercase letters for the same phenolic fraction indicate significant differences (*p* < 0.05) among treatments.

**Figure 3 antioxidants-11-00337-f003:**
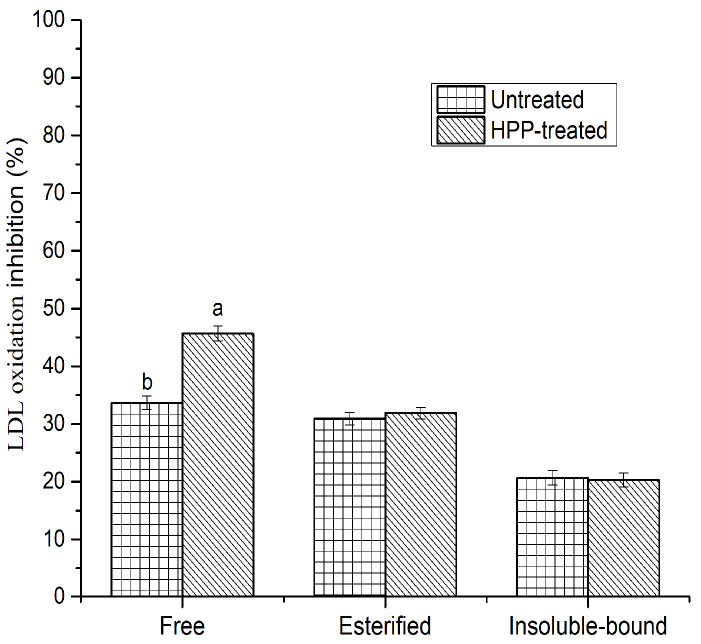
Inhibition against human LDL cholesterol oxidation by HPP-treated and untreated sea cucumbers. Different lowercase letters for the same phenolic fraction indicate significant differences (*p* < 0.05) among treatments.

**Figure 4 antioxidants-11-00337-f004:**
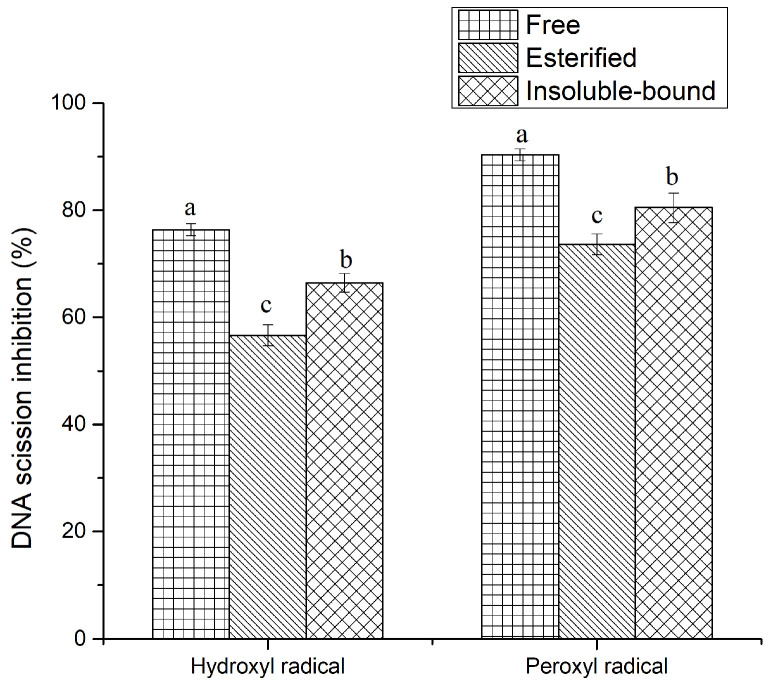
Inhibition of hydroxyl and peroxyl radical-induced DNA scission by HPP-treated sea cucumber internal organ phenolics. Different lowercase letters indicate significant differences (*p* < 0.05) among phenolic fractions.

**Figure 5 antioxidants-11-00337-f005:**
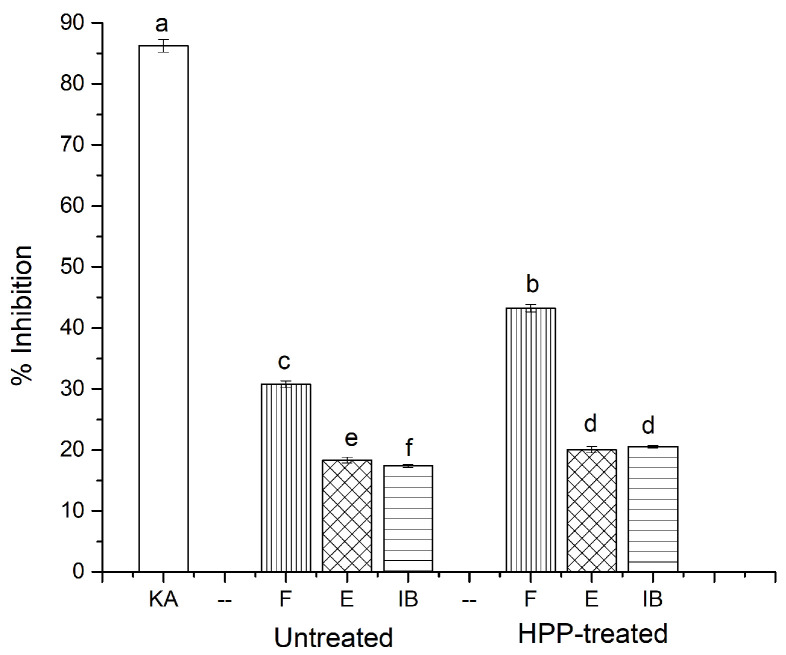
Anti-tyrosinase activity of untreated and HPP-treated sea cucumber. Different lowercase letters indicate significant differences (*p* < 0.05) among treatments. KA, kojic acid; F, free; E, esterified; IB, insoluble-bound.

**Figure 6 antioxidants-11-00337-f006:**
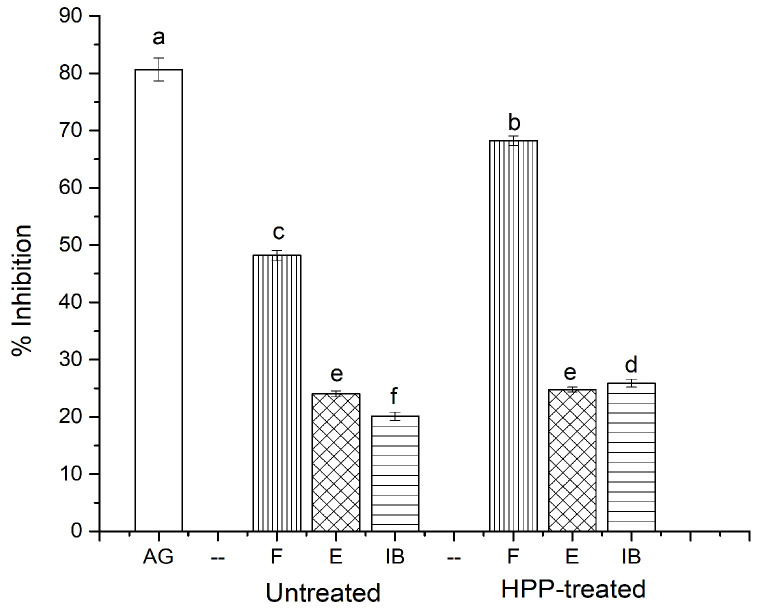
Antiglycation activity of HPP-treated and untreated sea cucumber. Different lowercase letters indicate significant differences (*p* < 0.05) among treatments. AG, aminoguanidine; F, free; E, esterified; IB, insoluble-bound.

**Table 1 antioxidants-11-00337-t001:** Antioxidant activity of HPP-treated and untreated sea cucumber internal organs.

Assays	Free		Esterified		Insoluble-Bound	
	Untreated	HPP-Treated	Untreated	HPP-Treated	Untreated	HPP-Treated
ARSA (mg TE/100 g)	589.18 ± 2.18 ^a^	565.43 ± 2.06 ^b^	89.67 ± 0.94 ^a^	76.54 ± 0.75 ^b^	109.03 ± 0.98 ^a^	94.76 ± 0.58 ^b^
DRSA (mg TE/100 g)	330.56 ± 1.68 ^b^	346.48 ± 1.06 ^a^	39.76 ± 0.52 ^b^	43.72 ± 0.75 ^a^	66.78 ± 0.76 ^b^	72.76 ± 0.58 ^a^
HRSA (mg TE/100 g)	598.33 ± 1.58	598.93 ± 1.05	131.89 ± 0.82	132.37 ± 0.75	226.76 ± 1.5 ^b^	254.3 ± 0.58 ^a^
MCA (mg EDTAE/100 g)	18.68 ± 0.28 ^b^	25.67 ± 1.05 ^a^	4.09 ± 0.2 ^b^	11.78 ± 0.75 ^a^	6.45 ± 0.42	7.27 ± 0.58

Different lowercase letters for the same phenolic fraction indicate significant differences (*p* < 0.05) among treatments. ARSA, ABTS radical scavenging activity; DRSA, DPPH radical scavenging activity; HRSA, hydroxyl radical scavenging activity; MCA, metal chelation activity; TE, Trolox equivalents; EDTAE, ethylenediaminetetraacetic acid equivalents.

**Table 3 antioxidants-11-00337-t003:** Quantification of phenolic compounds (mg/100 g) in HPP-treated and untreated internal organs.

C#	Compounds	[M − H]^−^ (*m/z*)	HPP-Treated			Untreated		
			F	E	IB	F	E	IB
**1**	*p*-Hydroxybenzoic acid	137	1.4 ± 0.03	0.49 ± 0.08		1.35 ± 0.03	0.46 ± 0.06	
**2**	Cinnamic acid	147	3.24 ± 0.28 *		1.54 ± 0.22	2.58 ± 0.15		1.51 ± 0.16
**3**	Protocatechuic acid	153	3.69 ± 0.24 *	1.39 ± 0.1 *	2.05 ± 0.18 *	2.68 ± 0.26	1.12 ± 0.08	1.46 ± 0.08
**4**	*p*-Coumaric acid	163	2.89 ± 0.08	1.38 ± 0.05 *	2.88 ± 0.2 *	2.81 ± 0.01	1.11 ± 0.1	1.8 ± 0.12
**5**	Vanillic acid	167	1.37 ± 0.12	0.73 ± 0.1	0.95 ± 0.06		0.7 ± 0.06	0.89 ± 0.1
**6**	Gallic acid	169	3.22 ± 0.32 *	0.89 ± 0.12	1.55 ± 0.15	2.48 ± 0.18	0.85 ± 0.1	1.51 ± 0.22
**7**	Caffeic acid	179			1.01 ± 0.09			1.02 ± 0.16
**8**	Homovanillic acid	181	1.03 ± 0.06					
**9**	Hydroxygallic acid	187	3.25 ± 0.34 *	0.53 ± 0.04	2.42 ± 0.18 *	2.24 ± 0.26	0.51 ± 0.08	1.89 ± 0.12
**10**	Isoferulic acid	193			0.76 ± 0.15			
**11**	Syringic acid	197	2.55 ± 0.1	0.34 ± 0.1	1.34 ± 0.05	2.5 ± 0.08	0.34 ± 0.05	1.3 ± 0.08
**12**	*p*-Coumaroyl glycolic acid	221		0.25 ± 0.1	0.66 ± 0.18			
**13**	Sinapinic acid	223	2.5 ± 0.18 *	0.57 ± 0.12		1.79 ± 0.2		
**14**	Ellagic acid	301	1.66 ± 0.15	0.54 ± 0.15	2.13 ± 0.28	1.61 ± 0.1	0.51 ± 0.12	2.1 ± 0.24
**15**	Chlorogenic acid	353	2.47 ± 0.18		3.06 ± 0.08 *	2.4 ± 0.05		2.46 ± 0.1
**16**	Caffeoyl glucoside	387	2.47 ± 0.12			2.41 ± 0.16		
**17**	Chicoric acid	473			0.73 ± 0.16			
**18**	Catechin	289	5.8 ± 0.58 *	1.2 ± 0.08 *	2.33 ± 0.32	4.38 ± 0.36	0.85 ± 0.1	2.33 ± 0.16
**19**	Quercetin	301	3.05 ± 0.32	0.73 ± 0.18	1.7 ± 0.2	3.02 ± 0.14	0.7 ± 0.06	1.66 ± 0.1
**20**	*p*-Hydroxybenzaldehyde	121	1.81 ± 0.08	0.72 ± 0.06	0.81 ± 0.18	1.75 ± 0.1	0.71 ± 0.12	0.75 ± 0.08
**21**	*p*-Hydroxycoumarin	161		0.8 ± 0.16				
**22**	Scopoletin	191	1.56 ± 0.16					
**23**	Leachianol F	471	0.63 ± 0.05					
Total			44.58	10.56	25.92	33.97	7.86	20.67
Total phenolic content			81.06			62.5		

All data represent the mean of triplicates. * Indicates significant differences (*p* < 0.05) for the same phenolic fraction among treatments. F, free; E, esterified; IB, insoluble-bound.

## Data Availability

Data are contained within the article.

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
