# Peer review of "Phenolic Compounds and Antioxidant Capacity of Sea Cucumber (*Cucumaria frondosa*) Processing Discards as Affected by High-Pressure Processing (HPP)"

_antioxidants, 2022, doi:10.3390/antiox11020337_

Round 1
Reviewer 1 Report
Phenolic compounds and antioxidant capacity of sea cucumber (Cucumaria frondosa) processing discards as affected by high-pressure processing (HPP) is very interesting and well written.
Presented manuscript is on good scientific level and represent a very high scientific value manuscript.
The summary. Authors give a short presentation of manuscript.
Introduction section.
The Introduction section includes all necessary information about examined objects and problems. Formatted aim and main hypotheses are good presented at the end of Introductions' section. The problem described in manuscript is a very new and represent a new food production and using of plants which are a completely not known or forgotten as a food.
Materials and method section
In the first sentence of sub section Material and methods Authors should describe how many sea cucumbers were used for experimental purposes. Please add this information.
Results
All data in Tables and Figures are good presented.
The discussion section presents a good comparison of the obtained results with other results available in the data basis.
Presented conclusions are corresponding with all information presented via Authors’ in manuscript text.
General opinion: After carefully manuscript reading, I think, that presented manuscript is a very valuable. In my opinion Manuscript should be accept with minor correction according my suggestion. Manuscript is perfect to Antioxidants journal.
Reviewer 2 Report
The possibility further exploit sea cocumber processing dischards for the extraction of antioxidants makes sea cocumber supply chain more sustainable under both economic and environamental aspects. The Authors clearly focused their reseach on the specific methodology that can be applied to extract such valuable compounds form internal organs of sea cocumbers. The methodology is well reported and results are well organized in graphs and tables. furthermore, the results have been discussed according to the current literature and the conclusions are supported by results.
I suggest only to check a double space int the following sentence "Moreover, due to HPP, solvents may easily penetrate into samples and enhance" pag 7, par 3.1.
Possibly, in the duscussion section, explain if this methodology can be applied to other species as well.
the mass transfer and permeability, thus improving extractio"
